# Electrical manipulation of telecom color centers in silicon

Aaron M. Day [1,4], Madison Sutula [2,4], Jonathan R. Dietz [1], Alexander Raun[1], Denis D. Sukachev[3], Mihir K. Bhaskar[3] & Evelyn L. Hu [1]✉

Silicon color centers have recently emerged as promising candidates for commercial quantum technology, yet their interaction with electric fields has yet to be investigated. In this paper, we demonstrate electrical manipulation of telecom silicon color centers by implementing novel lateral electrical diodes with an integrated G center ensemble in a commercial silicon on insulator wafer. The ensemble optical response is characterized under application of a reverse-biased DC electric field, observing both 100% modulation of fluorescence signal, and wavelength redshift of approximately $1.24 \pm 0.08$ GHz/V above a threshold voltage. Finally, we use G center fluorescence to directly image the electric field distribution within the devices, obtaining insight into the spatial and voltage-dependent variation of the junction depletion region and the associated mediating effects on the ensemble. Strong correlation between emitter-field coupling and generated photocurrent is observed. Our demonstration enables electrical control and stabilization of semiconductor quantum emitters.

Silicon is a foundational material enabling applications across computation, electronics, and photonics. It is, therefore, intriguing to consider it as a host for quantum information processing applications. Although color centers in solids have emerged as a promising quantum memory platform, the most mature color center technologies[1–11] are hosted in materials that are difficult to fabricate, such as diamond and silicon carbide. Recently, progress in studying the G and T centers has renewed interest in using silicon color centers as quantum emitters[12–16]. Additionally, demonstrations of silicon color center nanophotonic integration[17–24] reveal the potential to leverage the long history of scalable device engineering in silicon to realize useful quantum technologies. Yet, a better understanding of material processing is needed to achieve high yield and reproducible formation of single G and T centers.

While most attention is devoted to integrating color centers with photonic structures to enhance defect readout, integration with electronic devices offers complementary benefits for defect control. Electronic control of color center performance has already been

observed in other platforms for linewidth-narrowing[8], Stark tuning[8,25,26], charge state manipulation[27,28], and readout[29]. Ultimately, electronic control of single color centers could permit the tuning and stabilization necessary to produce indistinguishable spin-photon interfaces. Initially, however, it is useful to study the interaction of an ensemble of emitters with an electronic structure because it enables robust mapping of the device's electronic characteristic via the ensemble's local coupling to the device's electric field. Such ensemble-level device characterization is critical to understand the typical performance of defects in future devices used for single defect control.

Critically, the electric field interaction of the silicon G center is yet to be characterized. Correspondingly, electronic devices can provide important insights on the defect and charge noise environment and their effects on the optical and spin coherence of the integrated color centers. In this article, we describe such an electrical platform: we investigate the cryogenic optical response of a silicon color center to an applied electric field by integrating an ensemble of G centers with lateral $p^+$-$p$-$n^+$ diodes fabricated in silicon on insulator (SOI) (Fig. 1a).

[1]John A. Paulson School of Engineering and Applied Sciences, Harvard University, Cambridge, MA 02138, USA. [2]Department of Physics, Harvard University, Cambridge, MA 02138, USA. [3]AWS Center for Quantum Networking, Boston, MA 02135, USA. [4]These authors contributed equally: Aaron M. Day, Madison Sutula. ✉e-mail: ehu@seas.harvard.edu

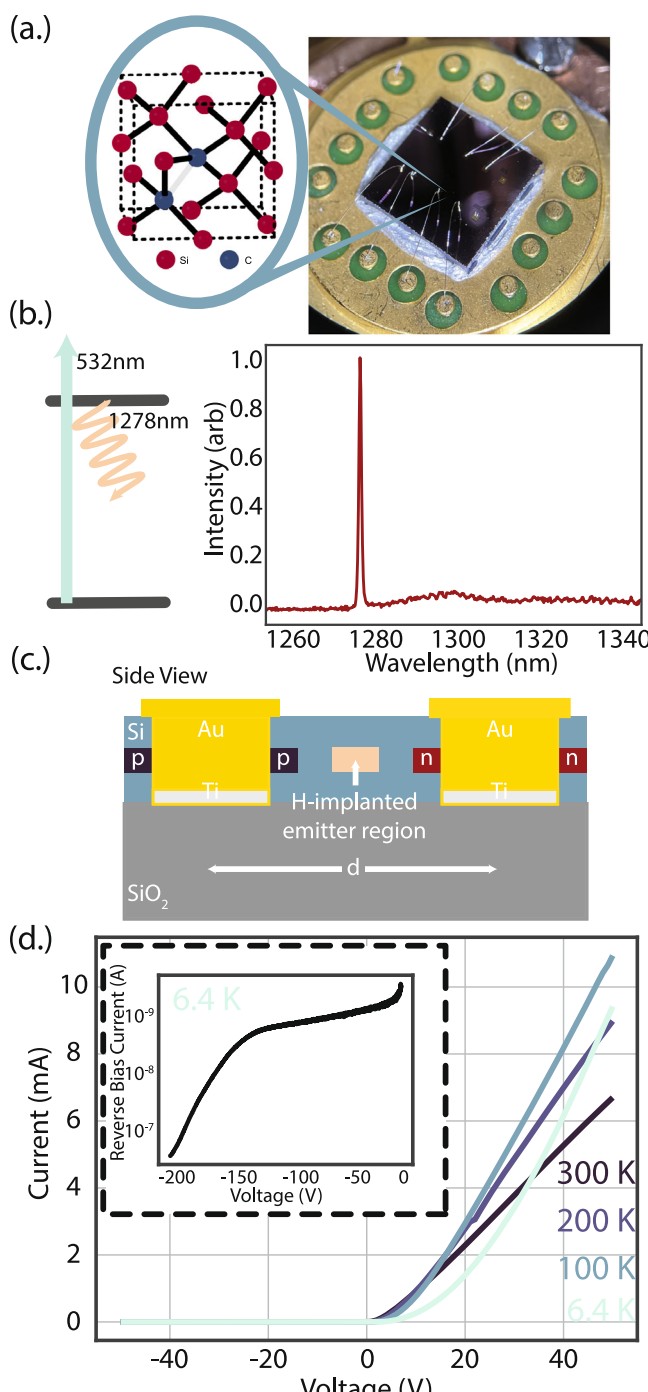

**Fig. 1 | Diode-integrated silicon G centers. a** Carbon-related silicon color centers are integrated into lateral p$^+$-p-n$^+$ junctions (diodes) fabricated in silicon on insulator and electrically driven by a wire-bonded 16-pin helium cryostat connector. **b** The color centers are optically excited by a 532 nm laser and fluoresce at 1278 nm in the telecommunication O-Band. **c** Side profile of fabricated diodes. P- and n-doping is achieved via ion implantation, and hydrogen is locally incorporated to selectively form G centers at the junction center. **d** Current-voltage (IV) curves of packaged diodes with integrated G center ensemble, measured as cryostat cools to base operating temperature of approximately 6 K. Inset shows low reverse bias leakage current, passing −0.5μA at -200 V.

The G center–comprised of two substitutional carbon atoms bound to an interstitial silicon atom–is an optically-active O-band emitter (Fig. 1b). Hydrogen implantation was found to be necessary for formation of G centers within our devices. By selectively implanting hydrogen by photoresist masking a wafer previously blanket-

implanted with carbon, it is possible to create an ensemble localized to the unmasked region in the center of a fabricated diode, where the depletion region forms under external bias. Above a spatially-dependent threshold voltage, the ensemble zero phonon line (ZPL) wavelength experiences a redshift up to 100 GHz at a rate of approximately 1.24 ± 0.08 GHz/V. Additionally, we observed the continuous reduction of the G center optical fluorescence with increasing reversed bias voltage, and at -210V the fluorescence was fully suppressed. Finally, we employ the observed emitter-field coupling to image the spatial distribution of the electric field within the junction.

The resultant spatial dependence of ZPL tuning and fluorescence extinction suggest these mechanisms could be attributed to a combination of the Stark effect of the electric field interacting with the dipole moments of the defects and the change in charge populations of the defects wrought by the Fermi level change of relative charge populations under band bending. Our method has broad applicability for future control in quantum networking experiments, and serves as a tool for probing fundamental color center behavior. Similarly, utilizing a color center ensemble enables precise determination of the local electrical environment experienced by the emitters across the PN junction. This approach is readily extensible to probe and control other color centers in silicon, and color centers in a wide range of semiconductor platforms, which are easily doped, such as silicon carbide.

## Results

### Lateral diodes with integrated G center ensemble

Our platform realizes a spatially-isolated G center ensemble maximally interacting with an electrical diode at a buried plane which is ultimately compatible with integrated silicon photonics. The maximum optical mode concentration of photonic crystal cavities in a 220 nm silicon layer would reside at 110 nm, thus we implement a design and fabrication strategy to support future hybrid electrical-optical coupling of semiconductor quantum emitters. To facilitate this, an industry standard 220 nm SOI wafer was utilized, with a dopant-defined diode embedded at a depth of 110 nm. Ion implantation combined with successive aligned optical lithography writes enabled masked localized incorporation of the p- and n- dopants, and the G center ensemble, at the desired depth (Fig. 1c). An etch-defined metallization strategy was employed to ensure robust electrical contact and performance at the dopant plane, and the device was packaged for cryogenic characterization. The current-voltage characteristics of the devices (Fig. 1d) do not degrade with temperature or masked hydrogen implantation, and they exhibit low leakage current under high reverse bias (Fig. 1d inset).

### Device design and fabrication

Lateral diodes are fabricated in commercial SOI (University Wafer, 220 nm Si on 2 $\mu$m buried oxide insulator, boron-doped, $\rho = 10 - 20\,\Omega$ cm, $\langle 100\rangle$orientation) to facilitate simultaneous cryogenic optical and electrical measurement of color centers. The starting substrate of the devices is lightly p-doped based on prior reports of emitter synthesis[15,19], though electrical performance would be improved in intrinsic material. The device design enables ease of optical access, variable junction width, and wafer scale–where hundreds of devices with swept parameters can be fabricated on a single commercial wafer defined via optical lithography. Further, the design co-locates the formed color centers and dopant-defined junction in the same spatial plane, improving emitter-field interaction. Device performance is validated with the COMSOL Multiphysics Semiconductor Module (see Supplementary Note 6). The full device design and fabrication is depicted in Fig. 2a. A top-down diagram of the relevant regions of the device is illustrated in Fig. 2b, accompanied by an optical image of the finished devices (Fig. 2c). The full details of device fabrication are given in Methods.

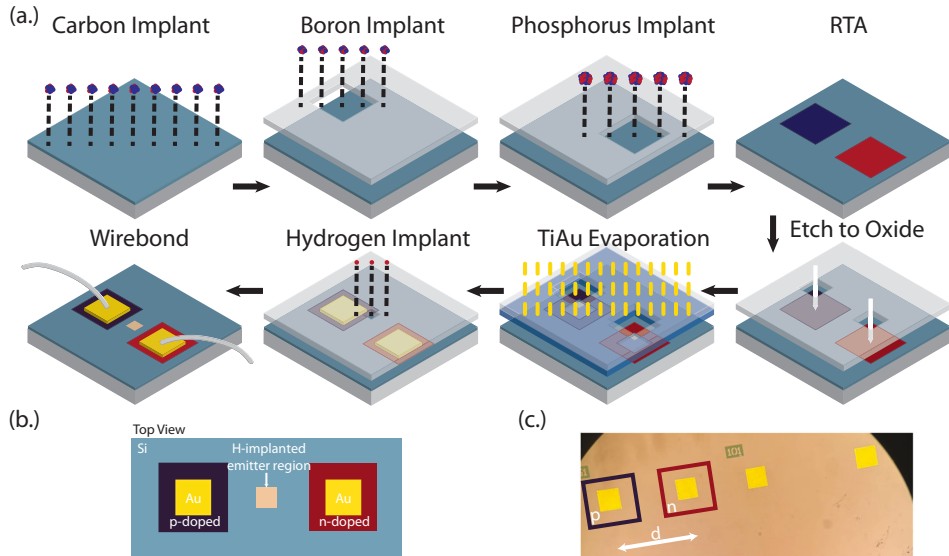

**Fig. 2 | Device design and fabrication. a** Fabrication process for realizing diode-integrated G centers in SOI. **b** Top view illustration of completed electrical device, and **c** associated optical micrograph depicting appearance of finished device, with notable regions denoted. The junction spacing d is varied across the wafer to enable a range of emitter-field coupling strengths.

## Ensemble incorporation

To form G centers within the diodes, we first implant Si with carbon ions then rapid thermal anneal at 1000°C to heal lattice damage. However, in contrast to some previous work[17–19,30], we did not observe G centers at this stage. Consistent with the findings of other works arguing proton irradiation facilitates incorporation of interstitial carbon into G centers[12,31], we investigated varied means of hydrogenation to complete the G center formation (see Supplementary Notes 7–8 in Supplementary information). Masked ion implantation of hydrogen was ultimately selected for the device-emitter integration to obtain a bright localized ensemble at the targeted depth where the electric field is strongest, with negligible degradation of electrical performance. Consistent with the findings of hydrogen's role in G center formation and stabilization, we found ensemble emission localized only to the implantation mask. Additionally, we found the G center to be unstable above 200°C Supplementary information, consistent with previous work[32], therefore requiring hydrogen incorporation to be the final fabrication step. Thus to ensure the diode fabrication was compatible with G center production, steps for both fabrication processes were interspersed.

## Electrical manipulation

Applying reverse-bias to a diode enables a number of advantageous controls to the environment of junction-integrated color centers. Under equilibrium conditions, the occupation of mid-gap defect states is determined by the positions of Fermi level to a defect's charge transition levels. By tuning the Fermi-level, either passively by introducing extrinsic dopants, intrinsic defects, or via surface functionalization, or actively through band bending in electronic structures, a defect's relative brightness may be modified by making a dark or bright charge state of the defect more statistically favored. Indeed, the Fermi level has been found to be a critical factor in a color center's optical activity[33–35]. Within the depletion region of a diode, the charge population is influenced by the locations of the quasi-Fermi levels, $E_{F,n}$ and $E_{F,p}$. In addition, the selective charge population of a defect may show greater stability, free from transitions from free electrons in the conduction band or free holes in the valence band. The internal electric field across the depletion region may also interact with the G centers to produce a Stark shift in the ZPL, as has been observed for other color centers[25,27,36]. Furthermore, reverse-bias driving involves low leakage current, which ensures minimal local heating. Therefore, electrical control of the junction-integrated color centers will modulate their photoluminescence spectra.

As such, we first characterized the optical response of the diode-integrated G center ensemble localized to the hydrogen implantation aperture under the application of a reverse-bias DC electric field. As a calibration of the local temperature, the silicon free-exciton was characterized under the same conditions. To gain a broader understanding of the spatial variation of the ensemble optical response, we monitored both photoluminescence and photo-induced current for different values of reverse bias voltage.

The G center ensemble response to a reverse-biased DC electric field is shown in Fig. 3a-b. Photoluminescence was measured while sweeping the reverse bias in 10 V intervals from 0 to -210 V (Fig. 3a). The fluorescence intensity of the ensemble was reduced as the reverse bias increased, until the signal dropped below the noise floor of the measurement (Fig. 3b). A 100 GHz redshift at a rate of approximately 1.24 ± 0.08 GHz/V was observed in the G center ZPL above a spatially-dependent threshold voltage. Additionally, the ensemble linewidth broadened as the center wavelength redshifted (see Supplementary Note 3A Supplementary information). In part, there is a $\sqrt{V}$ dependence of the electric field until the region is fully depleted[37]. The direction of the electric field across the ensemble selectively interacts with some portion of the G-center population, and depending on the direction of the Stark shifts, may broaden the overall luminescence peak observed. Future experiments with single emitters may help to elucidate the exact mechanisms at play.

A reverse biased current of -0.5 μA was passed at -200 V – corresponding to an applied power of 100 μW spread over a 103 μm junction gap. The device maintained low leakage current at high reverse bias, thus local heating is unlikely to be the source of the observed G center broadening, shifting, and modulation. To illustrate this point, the evolution of the silicon free-exciton line was investigated under the same bias conditions (Fig. 3c). The silicon free-exciton is suppressed at elevated temperature and thus served as a probe of local junction heating. The exciton photoluminescence (PL) measured from 0 to -210 V is shown in Fig. 3c. The exciton luminescence was not modified under reverse bias, consistent with the absence of significant

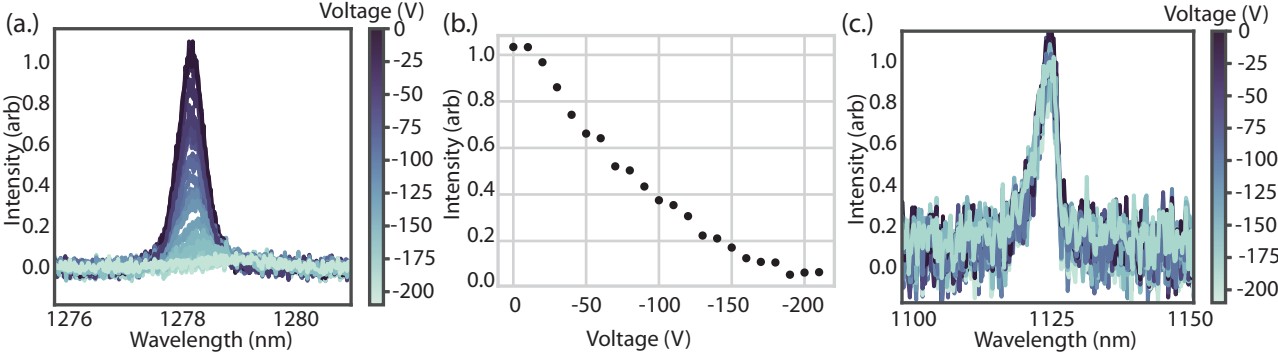

**Fig. 3 | Reverse-bias electrical manipulation. a** Optical response of G center ZPL under application of a DC electric field reverse biased from 0 to -210 V. Intensity continually decreases with increased reverse bias while the center wavelength redshifts by approximately 100 GHz. **b** Analysis of ZPL modulation ratio as a function of reverse bias, yielding 100% modulation at -210 V. Normalized to ZPL intensity at zero bias. **c** Preservation of silicon free-exciton under reverse bias. Unlike the G center, the exciton intensity, center wavelength, and linewidth are not correlated to increased reverse bias, indicating thermal effects are not a significant source of the observed G center behavior.

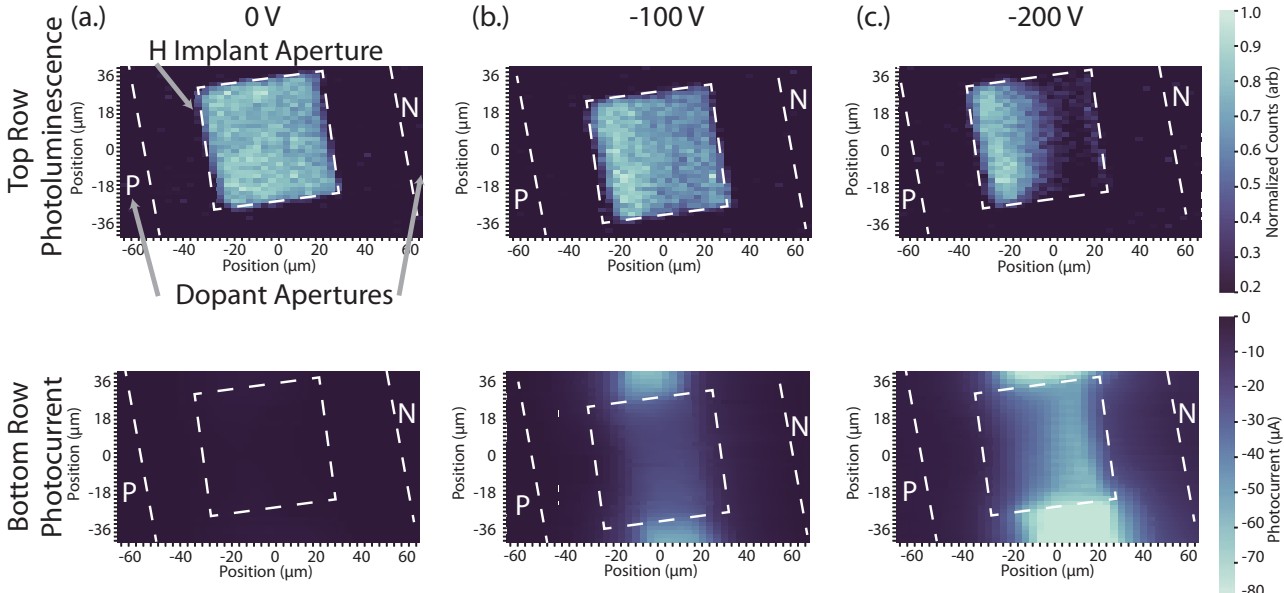

**Fig. 4 | Spatial distribution of emitter-field interaction.** Confocal maps of diode under reverse bias, depicting ensemble photoluminescence (top row) and measured junction photocurrent (bottom row) at (**a**) 0 V, **b** -100 V, and **c** -200 V. The junction depletion region distribution is illustrated via the ZPL optical intensity modulation spreading across the ensemble beginning from the region closest to the n-type contact, with strong agreement found by correlating the optical response with injected photocurrent measured in the junction. PL color bar noise floor is set to 0.2 due to 1 s integration. Modulation ratio is obtained from normalizing to mean ZPL intensity at zero bias from (**a**) top.

heating–with fluctuations attributable to noise in the experiment. We note that the asymmetry in the exciton stems from not fully resolved peaks associated with the transverse optical (TO) and longitudinal optical (LO) bands near 1130 nm[38]. These results of the G center and silicon free-exciton are contrasted with the behavior under the application of a high power forward bias in Supplementary Note 4 Supplementary information.

Capturing the distribution of emitter optical response across the junction can aide in characterizing the nature of the emitter-field interaction. The creation of an electric field across a depleted region is achieved simply by making the p-contact increasingly negative relative to the n-contact; the consequences should be evident in the photoluminescence profile across the junction. Ideally, given the lateral geometry of our diodes, we would expect a uniform increase in the depletion layer width with increased reverse bias. However, the central hydrogen-implanted region may influence the Fermi energy in that region, as well as the linearity of the reverse bias field between the

diodes. Photoexcitation of the reverse-biased junction will produce a photocurrent. This provides another way of mapping the depletion region, and hence a means of cross-correlating the spatial information given by the photoluminescence response of the defect ensemble. To this end, the spatial distribution of the electric-field coupling to the G center ensemble is imaged (Fig. 4 top row), and correlated with the associated optically-generated photocurrent (Fig. 4 bottom row) of the diode under 0-bias (Fig. 4a), -100 V (Fig. 4b), and -200 V (Fig. 4c).

At 0 V, the localization of the G center ensemble is clear (Fig. 4a top). G center PL is only observed in the 50 × 50 μm aperture at the center of the diode through which hydrogen was implanted. As expected under zero bias, the measured photocurrent is negligible (Fig. 4a bottom).

The confocal scan was repeated across the junction at a reverse bias of -100 V (Fig. 4b). At -100 V, the optical intensity modulation ratio of the G centers was spatially dependant, with the emitters in the portion of the hydrogen-implant aperture closest to the n-contact

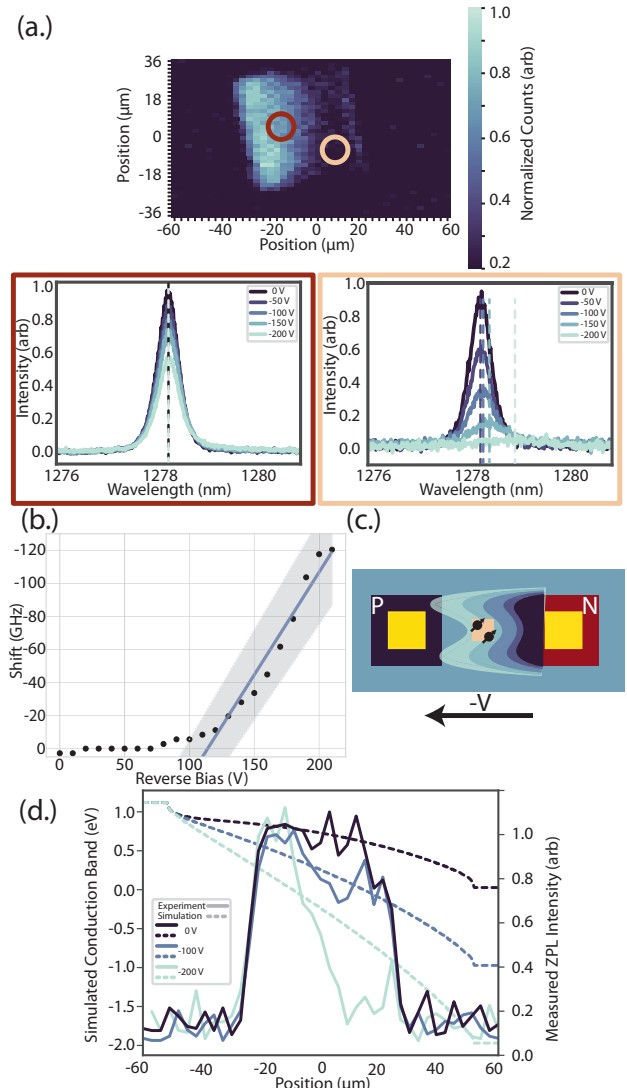

**Fig. 5 | Variation in G center optical response across junction. a** The ensemble response is analyzed across the confocal PL map of Fig. 4c from 0 and to -200 V. Close to the n-contact (brown), the ensemble experiences complete optical modulation and wavelength tuning. Tuning depicted as distance between dotted lines (full data set is given in Fig. 3). At the junction center (red), the ensemble experiences partial optical modulation with no wavelength tuning. **b** Ensemble center-wavelength redshift versus applied reverse bias of 1.24 ± 0.08 GHz/V, obtained from Gaussian fits of Fig. 3a data (details in Supplementary Note 2). Linear fit given with one standard deviation. **c** Illustration of observed reverse-bias redshift. The junction depletion (wavy curves) reaches the ensemble (center orange square with black emitters) at a sufficiently large threshold voltage (blue to teal colors), resulting in wavelength tuning. **d** Spatial correlation between simulated band bending (dashed lines) and 1D slice of the experimentally measured ZPL intensity (solid lines) at increasing reverse bias.

showing 40% greater suppression in response to the applied electric field than those nearest the p-contact. Interestingly, when comparing the confocal PL (top) with the associated confocally-excited photo-current measured in the device (bottom), the presence of the ensemble—and thus hydrogen—decreased the current passage across the junction, as the regions within the junction above and below the implant aperture demonstrated higher photo-responsivity. We believe that this may be due to (a) a different reverse-bias profile in those regions, as well as (b) the H-implantation providing trap states for the photo-generated electrons and holes[39], resulting in a stronger photo-current intensity outside the aperture.

Finally, the confocal PL and photocurrent spatial scan was repe-ated at a reverse bias of -200 V (Fig. 4c). Closer to the n-type contact, 100% modulation of the G center fluorescence is observed (Fig. 4c top). Conversely, closer to the p-contact, G centers are minimally suppressed. Furthermore, the associated confocally-excited photo-current (Fig. 4c bottom) follows the same spatial pattern. Optically-generated photocurrent measured in the junction was maximum in the region where emission is maximally modulated, confirming that the region of greatest depletion corresponds with the strongest emitter interaction. As the strength of the reverse bias field increased, the spatial extent of the ensembles experiencing the greatest optical modulation spreads from the n-contact toward the p-contact as elec-trons and holes are swept toward their respective n- and p- contacts (Fig. 4 a-c).

Furthermore, although partial optical modulation was observed at the center of the junction, wavelength tuning was not (Fig. 5a red). This finding is consistent with those reported experimentally[8], and theoretically[37], where at reverse bias voltages below a critical value, the size of the depletion region is less than the width of the junction. In[8], the threshold voltage to observe the Stark effect of single divacancies in 4H-silicon carbide positioned at different spatial planes of a vertical diode depended on the position of the emitter in the junction. Here, we extend this argument by directly imaging the spatial dependence of the entire diode depletion region. Above a spatially-dependent threshold voltage where the junction depletion reaches the ensem-ble (Fig. 5a brown), a continual redshift of approximately 1.24 ± 0.08 GHz/V is observed (Fig. 5b). However tens of microns away, where the junction depletion has not yet overlapped with the ensemble, no wavelength tuning is experienced (Fig. 5a red). The non-uniform field and the presence of electrical noise in the form of excess charge obfuscates the differential polarizability of the G center[18] and the exact nature of Stark shift tuning. However future work using single emitters in an undoped I layer of a PIN diode would enable this estimation to compare with theoretical predictions of the permanent dipole moment[40], as the precise Stark shift rate would be clearly captured by a single emitter and the lack of residual dopants would result in improved electric field uniformity. Wavelength-tuning is only observed in regions that also exhibit strong photocurrent, indicating the presence of the junction depletion region and large local electric field. This observation is conceptually illustrated in Fig. 5c, where the junction depletion (wavy lines)–as confirmed via confocally-excited photocurrrent–and corresponding emitter tuning spreads from the n contact toward the p contact with increased reverse bias (lighter shades). These results suggest the Stark effect could be responsible for the observed emitter red-shift. The boron dopants in these areas within the junction are sufficiently ionized such that an electric field can build up to yield Stark-shifted G centers.

Finally, G center optical intensity is modulated both within and outside of the depletion region under increasing reverse bias. This observation could be explained by considering the effects of defect charge state modification via Fermi-level engineering. The trend and spatial relation of predicted band bending and ensemble brightness is well correlated (Fig. 5d), indicating depletion of the optically active charge state at increasing reverse bias. G-centers can be ionized to non-emissive charge states as the Fermi level is tuned under external bias[41,42]. Our observations are consistent with this explanation: as we increase the reverse bias across the junction, emitters are probabil-istically ionized to a dark state as a function of the resultant band bending.

The center wavelength and nominal (zero-bias) brightness of the ensemble returns upon termination of the diode bias–both in the forward and reverse bias regimes–thus the charge state of the ensemble was not permanently altered by the measurements. Addi-tionally, there is no observed time-delay in the restoration of the emitter optical properties, though more sensitive future time-domain

measurements would provide precise detail on the transient nature of the response.

## Discussion

We probed the coupling of a telecommunication-band silicon color center to DC electric fields by integrating G centers into diodes while retaining optical access. We then utilized the electrical manipulation of the ensemble to image the electric-field distribution within the diode, capturing the spatial evolution of the junction depletion region across varied reverse-bias voltages. Within the junction depletion region the ZPL both diminished in intensity and redshifted by approximately 100 GHz at a rate of $1.24 \pm 0.08$ GHz/V above a threshold voltage, whereas only modulation of the ZPL fluorescence intensity is observed outside of the depletion region. These findings suggest distinct emitter-field couplings are exhibited—with a spatial dependence across the junction—where charge state modulation and Stark effect could explain the observed phenomena. Furthermore, we find that hydrogen plays a critical role in the ability to observe G centers in our devices. To this end, future work will continue to elucidate the specific mechanisms involved in G center formation and stabilization, both via hydrogenation and electrical control. Lastly, we note that the ability we have shown to convert a defect from an optically bright state to a dark one is similarly possible to occur in the reverse—where Fermi engineering via applied reverse bias may favorably populate a bright state from a dark one. The equilibrium Fermi level of the substrate will dictate the charge population of the defect, which is given by the background doping. In this demonstration with the selected host wafer, G centers were optically active at equilibrium, and therefore increased reverse bias may have depopulated the optically active charge state.

These devices provide a tool for electrically manipulating color centers with broad applicability to both other silicon color centers, and color centers in other semiconductor platforms. These findings using an ensemble of color centers to illustrate the spatial distribution of emitter-field coupling in the junction will motivate and inform the design of electrical devices to optimally couple to a single emitter. It would be of particular interest to observe the response of silicon T centers to electrical tuning via diode, as T centers possess a coherent spin-photon interface[15], and are reported to follow a similar synthesis procedure as was implemented here. Furthermore, our demonstration of the direct visualization of electric field dynamics in a semiconductor—optically mapping a DC electric field in-situ—has application in quantum sensing of electric fields[43]. Finally, our lateral diode design at a buried plane of 110 nm is compatible with photonic crystal cavity integration[44,45], where future work intends to enable simultaneous electrical tuning, stabilization, and control of cavity-enhanced quantum emitters.

## Methods

### Fabrication

All carbon, hydrogen, boron, and phosphorus ion implantation was performed at INNOViON Corporation. Ion implantation energies are determined using Stopping Range of Ions in Matter (SRIM) calculations Supplementary information, targeting a depth of approximately 110 nm for each ion. Dopant densities are selected to obtain an acceptor/donor concentration of $1 \times 10^{19}/cm^3$ at the desired depth, as this order magnitude is typical of electrical devices in silicon. Further, overlapping the maximum dopant concentration depth with the etch-defined metalization ensures transmissive metal-semiconductor interface for ohmic contact. Each implantation was performed at a 7° tilt. All masked implantation utilized optical lithography in the positive photoresist mask S1813 at a fluence of 250 mJ and wavelength of 375 nm using a Heidelburg Maskless Aligner 150. The resist was pre-baked at 115 °C for 3 minutes, and developed for 70 seconds in TMAH-based CD-26. Every photoresist mask was stripped with a 500 W $O_2$ plasma,

and the Ti-Au resist-on-liftoff mask was stripped with a 12 hr soak in remover PG at 80 °C.

First, an unmasked bulk wafer fragment is implanted with $7 \times 10^{13}/cm^2$ $^{12}C$ ions at an energy of 38 keV. Next, $500 \times 500\ \mu m$ apertures are written in a photoresist mask with optical lithography, and $1 \times 10^{14}/cm^2$ $^{11}B$ ions are implanted at an energy of 29 keV through the apertures to define localized p-doped islands. After resist stripping, n-doped islands are generated by implanting $1 \times 10^{14}/cm^2$ $^{31}P$ ions at an energy of 80 keV through offset $500 \times 500\ \mu m$ apertures again defined with optical lithography. The spacing between the p- and n-doped apertures (Fig. 2b) is swept across the wafer to vary the strength of the junction electric field for a given voltage. To both heal the crystal lattice and incorporate the dopants substitutionally in the silicon lattice[32], a rapid thermal anneal (RTA) is performed at 1000 °C for 20 seconds in an argon environment after stripping the resist.

Next, electrical contacts are generated by first writing $250 \times 250\ \mu m$ apertures in a new resist mask positioned such that each opening was aligned to the center of the implanted dopant islands. Using $SF_6$ and $C_4F_8$ chemistry in a reactive ion etching chamber, the exposed windows are then etched down 220 nm to the oxide to ensure optimal overlap of the metal contacts with the implanted dopants. Following definition of a new $300 \times 300\ \mu m$ aperture mask of photoresist on lift-off (S1813 on LOR3A), also aligned to the center of the implanted dopant islands, a thin film of 300 nm gold on a 30 nm titanium adhesion layer (Ti-Au) is deposited via electron beam evaporation.

To complete the incorporation of G center ensembles, hydrogen is implanted through a window at the center of each junction (Fig. 2b). $7 \times 10^{13}/cm^2$ H ions were implanted at an energy of 9 keV through $50 \times 50\ \mu m$ apertures in a final resist mask, forming an ensemble of diode-integrated G centers. The wafer fragment was subsequently diced into $6 \times 6$ mm samples that were integrated into a 16-pin electrically wired cryogenic cold-finger and wire-bonded for external driving (Fig. 1a).

### Experimental Setup

Experiments are performed in a home-built confocal microscope using a Mitutoyo $100 \times 0.5$ NA Near-IR objective. G centers are optically excited using an off-resonant 532 nm diode-pumped solid-state laser, and junctions are biased using a $\pm 210V$ Keithley 2400 source meter. Simultaneous optical and electrical measurements are enabled in a Janis ST-500 continuous-flow Helium-cooled cryostat with a 16-pin mapped electrical feed-through wire-bonded to the diodes. The system achieves a base temperature of roughly 6 K. Photoluminescence of the diode-integrated color centers is measured on an Acton Spectra Pro 2750 spectrograph with a Princeton Instruments OMA:V indium-gallium-arsenide nitrogen-cooled photodiode array detector. Raman spectroscopy is performed in a LabRAM Evolution Horiba multi-line room-temperature confocal Raman spectrometer using 532 nm laser excitation.

### Confocally-excited photocurrent

Here we provide additional information on the measurement performed to generate the data in Fig. 4. The excitation laser beam was scanned across the diode using a fast steering mirror. During the laser scan, at every scan point, a triggering pulse was sent to the spectrometer and to the source-meter applying reverse bias to simultaneously record the photoluminescence spectrum and the optically injected photocurrent in a given confocal spot. This enabled us to precisely map out the spatial correlation of ensemble photoluminescence and current in the diode simultaneously.

## Data availability

The data that support the findings of the work are presented in the Article and Supplementary Information. Source data are available from the corresponding author upon request.

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

## Acknowledgements

This work was supported by AWS Center for Quantum Networking and the Harvard Quantum Initiative. Portions of this work were performed at the Harvard University Center for Nanoscale Systems (CNS); a member of the National Nanotechnology Coordinated Infrastructure Network (NNCI), which is supported by the National Science Foundation under NSF award no. ECCS-2025158. M.S. acknowledges funding from a NASA Space Technology Graduate Research Fellowship.

## Author contributions

Methodology: A.M.D., M.S., J.R.D., D.D.S., M.K.B., E.L.H; Fabrication: A.M.D., A.R.; Measurement: A.M.D., J.R.D.; Analysis: A.M.D., M.S., J.R.D., D.D.S., M.K.B., E.L.H; Advising: D.D.S., M.K.B., E.L.H; Manuscript Preparation: All authors.

## Competing interests

A.M.D., M.S., J.R.D., D.D.S., M.K.B., E.L.H filed a patent application (US Provisional Application No. 63/636,525) related to the technology presented in this work. A.R. declares no competing interests.
