## [Peer Review File · Nature Communications]

REVIEWER COMMENTS

Reviewer #1 (Remarks to the Author):

In the manuscript, the emission wavelength and intensity of G center in silicon was electrically manipulated using a lateral PIN diodes. By doing correlated PL and photocurrent imaging of the whole diode, band bending and stark effect was proposed to be responsible for the intensity decrease and wavelength redshift, respectively, providing a useful tool for manipulating color centers in both silicon and other solids. The manuscript could be published, after addressing the following issues.

- (1) A schematic illustration for the band alignment in diode is suggested to be given, which would be very helpful for readers to understand the band bending.
- (2) Figure 5b needs to be well elaborated in main text for easy understanding. E.g. the wavy curves for electric fields, deep/light colors for different depletion degree.
- (3) Confocal is an optical method, ordinarily in scanning PL or Raman, for improving z-axis resolution. What does "confocal photocurrent" mean?
- (4) Figure 1b was wrongly referred in main text, and was actually not mentioned and discussed in main text.
- (5) The authors attributed the emission redshift to band bending. Why is redshift not blueshift? Some calculations would improve the manuscript.
- (6) In what way the field-emitter interaction decreased the PL intensity? Time resolved PL might give more details.
- (6) Figure 4, why obvious photocurrents appeared on the top and bottom sides of the aperture, not on the left-top, left-bottom, right-top, right-bottom?
- (7) Figure 4, one PL color bar on the row and one photocurrent color bar on the bottom row would be okay, since the color bars are the same for the top and the bottom images, respectively.
- (8) why the PL peaks of free-excitons in Figure 3c appeared asymmetric?

Reviewer #2 (Remarks to the Author):

The manuscript reported an experimental study of dependence of cryogenic optical response of G center ensemble on an applied electric field by fabricating lateral electrical diodes in a commercial silicon on insulator wafer. The ensemble ZPL experiences a redshift up to 100 GHz at a rate of approximately 1.4 GHz/V above the observed threshold voltage, and 100% extinction of the fluorescence intensity is observed. In addition, this study uses G center fluorescence to directly image the electric field distribution within the devices.

I consider that this study demonstrates electrical manipulation of G centers, which is timely and significant for the applications of G centers. This study provides a practical method for electrical control in quantum networking experiments. However, the paper also has some shortcomings, which should be revised before the publication.

1.The authors conclude that the electric field-dependent ZPL redshift rate of G centers is approximately 1.4 GHz/V, as shown in Fig. S1. I think it is a main conclusion, and it should be added to the main text and preferably with an error bar. Besides, it is clear that the results are poorly fitted and not convincing. Authors explain as the distribution of dipole orientations of individual emitters in the ensemble. Nowadays the study of single G center has been widespread [Nat. Commun. 13, 7683 (2022), Nat. Commun. 14, 2380 (2023), Nat. Commun. 14, 3321 (2023)], so I suggest the authors repeat the experiments with single emitters if possible.

2.The authors' conclusions about the optical response of G centers are expressed in terms of voltage. I consider the electric field plays a dominant role, and hope authors can express the relevant conclusions by the electric field.

3.I wonder whether this electrical manipulation is reversible.

Reviewer #3 (Remarks to the Author):

The authors report electrical tuning of the optical properties of the ensemble G centers in Si diode devices. The wavelength tuning is of vital importance for the quantum information technologies. They fabricated the G centers with hydrogen implantation after carbon implantation. The fluorescence at 1278 nm appears only at the region treated with hydrogen implantation, demonstrating the position selectivity for the G center fabrication. The electrical tuning of the emission wavelength and intensity is reported for the G centers. However, I think the present results are insufficient for the publication in Nature Communications.

1. Although the wavelength tuning is demonstrated, the intensity simultaneously decreases. This fact means that the formation of the depletion layer under voltage application makes the G centers unstable. Therefore, this technique is hard to directly use for the control of the G centers for quantum applications.

2. The wavelength shift with 1.4 GHz/V seems to be large enough to obtain indistinguishable photons. The dependence of the tuning on the voltage (or electric field) should be more discussed in terms of the effect of the atomic symmetry of the G center and electric dipole moment. Why isn't a single G center is utilized in this study to clarify these points?

3. The mechanism of the decrease in the intensity should be discussed. Is the charge state changed in the depletion layer, or another reason?

Color code:

- blue - Author opening statements
- black - reviewer comment
- red - author response to reviewer comment

Revisions to the main text are made in red.

Summary of Response

We appreciate the reviewers' thoughtful feedback on our manuscript "Electrical Manipulation of Telecom Color Centers in Silicon" and we are happy to provide modifications and clarifications to render our work acceptable for publication. We are glad to see the reviewers' positive reception of our findings, stating that our study provides "a useful tool for manipulating color centers in both silicon and other solids. The manuscript could be published, after addressing the following issues" (Reviewer 1), and [our work] "is timely and significant for the applications of G centers. This study provides a practical method for electrical control in quantum networking experiments." (Reviewer 2).

To preface our detailed response to reviewers' comments, we wish to make a general statement regarding what we believe is the principal value of this work. We regret that this was not more clearly evident in the original manuscript, and have rewritten the introduction to rectify this. In summary, we present a new design and method of co-locating an ensemble of color centers in a dopant-defined lateral electrical diode. This enables us to precisely characterize the local electrical environment experienced by the emitters, which has not been previously visualized. Our novel device geometry permits us to simultaneously measure the emitter response, providing new insight into defect physics and the potential to use our diode geometry for electrical control of quantum emitters.

As noted by the reviewers, the electronic control of *single* color centers would make possible the tuning necessary to produce indistinguishable color centers—critical for the implementation of quantum information systems. Indeed this work provides an important step toward achieving such tuning. But beyond this one goal—and perhaps of even greater importance in the initial characterization of candidate color centers—the electrical measurements described in this work as applied to *ensembles of color centers* enables a greater understanding of the formation of color centers, provides a more precise spatial mapping of the optical response of color centers relative to clearly evident regions of charge depletion and hence electric field, and offers a robust sampling of color center behavior in those regions. The particular geometry of our PN diodes produces electric fields across a planar region of color centers, generating direct images of a PN junction depletion region where color center luminescence is cross-correlated with photo-induced current that is a signature of the reverse-bias diode depletion. Thus our study provides complementary information to studies

of single G centers that is critical to understand the possibilities and limitations of forming, stabilizing, and controlling color centers in silicon as qubits.

Here we provide a summary of changes made in response to reviewer comments, and in following sections we delineate a point by point reply to each reviewer comment and provide specific action taken to address the request. The accompanying revised manuscript indicates updates with red text.

- We added a figure with new simulation and experimental data to correlate the spatial mapping of the measured fluorescence reduction with the simulated band bending in the diode (Fig. 5d). We find good agreement with the trend in and spatial occurrence of predicted band bending and altered ensemble photoluminescence—further confirming our findings.
- We improved the methodology used to fit the ensemble Stark tuning red-shift, added error bounds to the fit, and moved the figure to the main text (Fig. 5b).
- We added explanatory discussions throughout the main text, as requested by reviewers. (Meaning of confocal photocurrent, restoration of the nominal charge state, etc.)
- We added extensive discussion throughout the text to the interpretation of our findings in relation to the proposed mechanism of their effect. We elaborate on the process of Fermi level tuning via application of a reverse bias to modify the charge state of the defects in the ensemble, and on the Stark effect from local electric fields.
- We improved figure clarity as suggested by reviewers.

0.1 Reviewer #1 (Remarks to the Author):

In the manuscript, the emission wavelength and intensity of G center in silicon was electrically manipulated using a lateral PIN diodes. By doing correlated PL and photocurrent imaging of the whole diode, band bending and stark effect was proposed to be responsible for the intensity decrease and wavelength redshift, respectively, providing a useful tool for manipulating color centers in both silicon and other solids. The manuscript could be published, after addressing the following issues. **Reply: We thank the reviewer for their comments, and for recognizing the utility and applicability of our demonstrated technique. We address all of their suggestions in our point by point response below.**

- A schematic illustration for the band alignment in diode is suggested to be given, which would be very helpful for readers to understand the band bending.
- **Reply: We thank the reviewer for this excellent suggestion to add a schematic illustration of the band bending in the diode at the experimentally measured region. We strongly agree that this suggestion will further strengthen our findings, and support the reader in understanding their implications.**
- **Action Taken: We have created a new figure which compares the simulated band bending (dotted lines) in the ensemble region against an x-dimension slice of the measured confocal photoluminescence (solid lines). With this new figure analyzing simulated device function versus measured photoluminescence, the correlation between band bending and reduction in emitter fluorescence intensity is readily conveyed.**

- Figure 5b needs to be well elaborated in main text for easy understanding. E.g. the wavy curves for electric fields, deep/light colors for different depletion degree.
- **Reply: We thank the reviewer for providing opportunities to improve the understanding of our figures.**

- Action Taken: We have added the accompanying text below to elaborate on the details of the illustration. In addition, we added new data to Figure 5 to further convey the discussed ideas.

... junction depletion (wavy lines)—as confirmed via confocally-excited photocurrent—and corresponding emitter tuning spreads from the N contact toward the P contact with increased reverse bias (lighter shades). These results suggest the Stark effect could be responsible for the observed emitter red-shift.

- Confocal is an optical method, ordinarily in scanning PL or Raman, for improving z-axis resolution. What does “confocal photocurrent” mean?
- Reply: We thank the reviewer for this clarifying question. To address their question we added a paragraph to the main text, see below:
- Action Taken: We added a new methods section "Confocally-Excited Photocurrent:"

C. Confocally-Excited Photocurrent

Here we provide additional information on the measurement performed to generate the data in Fig. 4. The excitation laser beam was scanned across the diode using a fast steering mirror. During the laser scan, at every scan point, a triggering pulse was sent to the spectrometer and to the source-meter applying reverse bias to

simultaneously record the photoluminescence spectrum and the optically-injected photocurrent in a given confocal spot. This enabled us to precisely map out the spatial correlation of ensemble photoluminescence and current in the diode.

We also replaced each instance of "confocal photocurrent" with "confocally-excited photocurrent" to improve clarity.

- Figure 1b was wrongly referred in main text, and was actually not mentioned and discussed in main text.
- Reply: We thank the reviewer for this note. Figure 1b depicts the details of the G center’s optical activity, and we highlight its reference in the main text below.

In this article, we investigate the cryogenic optical response of a silicon color center to an applied electric field by integrating an ensemble of G centers with lateral p⁺-p-n⁺ diodes fabricated in silicon on insulator (SOI) (Fig. 1a). The G center—comprised of two substitutional carbon atoms bonded to an interstitial silicon atom—is an optically-active O-band emitter (Fig. 1b). Hydrogen implantation was found to be necessary for formation of G centers within our devices, and an ensemble is thereby localized to the middle of the diode junction by implanting hydrogen ions with a lithography-defined mask (Fig.

- The authors attributed the emission redshift to band bending. Why is redshift not blueshift? Some calculations would improve the manuscript.
- Reply: The reviewer has indeed posed an interesting question. These initial experiments have indicated that interactions of the G centers with the electric field, in fully-depleted regions of the diode, serve to redshift the wavelength and broaden the ensemble linewidth. In regions that are *not fully depleted*, there may be some charge-state modifications of the ensemble, but no shifts, nor linewidth changes (please see the revised Fig. 5). Until we better and more fully understand the symmetry and energy levels of the G center, and the methods of Stark interaction, we cannot carry out calculations, however we note that wavelength red-shift due to reverse bias is common [6, 4]. We have now clarified the caption for Fig. 5b.
- In what way the field-emitter interaction decreased the PL intensity? Time resolved PL might give more details.
- Reply: We thank the reviewer for this important question. We believe that within the depletion region, it is possible to alter charge populations of the inter-bandgap defect levels that are different from populations in non-depleted regions, which are affected by interactions with local electrons in the conduction band and holes in the valence band. It is known for other color center qubits that the charge value of the corresponding state can influence whether or not there is a radiative or non-radiative (dark) transition. An example is NV^- compared to NV^0 in diamond. The voltage-tuning, rather than making the G center *unstable*, may be placing it in a “dark” charge state. While this may not be desirable for the ultimate utilization of G centers in quantum systems, the important insight is that the G center *may have different charge states* that are “bright” or “dark”, and that deliberate Fermi level (charge population) is required to maintain the G centers in the preferred state. We agree with the reviewer that further time-resolved studies might provide the *lifetimes* of such desirable charge state defect energies within the depletion region; these would be most appropriate to carry out in future studies.
- Action Taken: The following text has been added to the main.

ing reverse bias. This observation could be explained by considering the effects of **defect charge state modification via Fermi level engineering**. The trend and spatial relation of predicted band bending and ensemble brightness is well correlated (Fig. 5d), indicating depletion of the optically active charge state at increasing reverse bias.

- Figure 4, why obvious photocurrents appeared on the top and bottom sides of the aperture, not on the left-top, left-bottom, right-top, right-bottom?
- Reply: The regions with the strongest photocurrent signatures are those with largest reverse bias voltage. In those regions, photo-excited electron-hole pairs are rapidly swept away by the strong reverse-bias fields. We believe that the reason for the stronger signatures above and below the aperture is that the H-implanted regions in the aperture may provide trap states for the photo-generated electrons and holes [3], thus diminishing the current. As suggested by the photoluminescence images, the left-most region, closest to the P-contact, is not fully depleted. Therefore the reverse bias voltage is less in that region, and so is the photocurrent. There is evidence for some photocurrent signature on the “right-top“ and “right-bottom“ portions.
- Action Taken: We have added the referenced citation, and the above discussion, to the main text.

shown the implant aperture demonstrated higher photo-responsivity. We believe that this may be due to (a) a different reverse-bias profile in those regions, as well as (b) the H-implantation providing trap states for the photo-generated electrons and holes [42], resulting in a stronger photo-current intensity outside the aperture.

- Figure 4, one PL color bar on the row and one photocurrent color bar on the bottom row would be okay, since the color bars are the same for the top and the bottom images, respectively.
- Reply: We thank the reviewer for this excellent suggestion to improve figure legibility.
- Action Taken: We have edited the figure as suggested to show one color bar per row.

- why the PL peaks of free-excitons in Figure 3c appeared asymmetric?
- Reply: We thank the reviewer for noting this detail. There are two free exciton peaks in silicon close to 1130 nm: one associated with the transverse optical (TO) band and another associated with the longitudinal optical (LO) band [5]. For the spectrometer grating used in the exciton measurements of $150 \text{ grooves mm}^{-1}$, the individual peaks—which are separated by about 2 nm—are not fully resolved.
- Action Taken: We have added this explanation to the main text.

noise in the experiment. We note that the asymmetry in the exciton stems from not fully resolved peaks associated with the transverse optical (TO) and longitudinal optical (LO) bands near 1130 nm [41]. These results of

0.2 Reviewer #2 (Remarks to the Author):

The manuscript reported an experimental study of dependence of cryogenic optical response of G center ensemble on an applied electric field by fabricating lateral electrical diodes in a commercial silicon on insulator wafer. The ensemble ZPL experiences a redshift up to 100 GHz at a rate of approximately 1.4 GHz/V above the observed threshold voltage, and 100% extinction of the fluorescence intensity is observed. In addition, this study uses G center fluorescence to directly image the electric field distribution within the devices.

I consider that this study demonstrates electrical manipulation of G centers, which is timely and significant for the applications of G centers. This study provides a practical method for electrical control in quantum networking experiments. However, the paper also has some shortcomings, which should be revised before the publication. **We thank the reviewer for their positive assessment of our work, and for recognizing that our demonstration is timely and significant. We are grateful for the clarifying questions and suggestions, which we are happy to address below.**

- The authors conclude that the electric field-dependent ZPL redshift rate of G centers is approximately 1.4 GHz/V, as shown in Fig. S1. I think it is a main conclusion, and it should be added to the main text and preferably with an error bar. Besides, it is clear that the results are poorly fitted and not convincing. Authors explain as the distribution of dipole orientations of individual emitters in the ensemble. Nowadays the study of single G center has been widespread [Nat. Commun. 13, 7683 (2022), Nat. Commun. 14, 2380 (2023), Nat. Commun. 14, 3321 (2023)], so I suggest the authors repeat the experiments with single emitters if possible.
- **Reply: We thank the reviewer for the suggestion to improve our fit of the Stark tuning redshift (completed and described below), and for the question about our decision to investigate ensembles rather than single emitters. We agree with the reviewer that the creation and investigation of single G centers is now widespread, and that a single emitter could provide a more precise determination of the emitter's dipole moment, symmetry, and Stark coupling. However, our choice to utilize an ensemble in this study is intentional. The goal of this work is to precisely characterize the electric field environment experienced by the G center so as to understand the interaction with a PN-junction. This is why we chose to use a dense ensemble which enabled the valuable insight gained regarding the spatial distribution and competing effects of the depletion region and band bending. Using the ensemble, we were able to differentiate between these two effects, and show how separate portions of the ensemble respond. Furthermore, we establish that a low-concentration P-doped substrate is not an advantageous selection for future devices which intend to Stark tune single emitters—due to the charge state depletion observed at low voltages. If we had performed this study with a single emitter at the center of our PN-junction, none of this information could have been gained.**

- Action Taken: We appreciate the suggestion to improve the fit of the observed Stark tuning, and to move the figure to the main text. We have now done so (new Fig. 5b), and have improved the fit in two ways. First, we analyzed the ensemble center wavelength by first fitting a Gaussian curve to the PL at each bias voltage and extracting the center of the fit (details of the specific methodology are now added to the supplement). Secondly, we fit a linear function (blue line) to the Gaussian fit-extracted center wavelength at each bias, and added bounds of one-standard deviation error (dashed lines bounding shaded region). We then updated the reported tuning rate accordingly ($1.24 \pm 0.08 \text{GHz/V}$) to include the error. Adding these two fitting methods has substantially improved the analysis, so we thank the reviewer for suggesting this change.

- The authors' conclusions about the optical response of G centers are expressed in terms of voltage. I consider the electric field plays a dominant role, and hope authors can express the relevant conclusions by the electric field.
- Reply: We thank the reviewer for this insightful suggestion. We completely agree that the relevant role in the emitter interaction is the local electric field, rather than applied voltage. However, as we discuss in the text, intrinsic dopants in the "I" layer of the junction inhibit typical field accumulation and prohibit complete junction depletion (as described in Anderson et al. Science 2019 [1], and Candido et al. PRX Quantum 2021 [2]), thereby rendering difficult the accurate determination of the local electric field experienced by the emitter. This is the reason for our spatially-dependent threshold voltage for Stark tuning, as was similarly observed in Anderson et al. Science 2019 [1].

We nonetheless can provide an estimate using the COMSOL junction simulations and the Lorentz local field approximation (useful in instances where dopants do not interfere with the measurement, as in Lukin et al. NPJ Quantum Information 2020). Comparing our experimentally measured Stark tuning rate per applied voltage (1.24GHz/V) to the field accumulation predicted by the COMSOL simulation (0.0098

(MV/m)/V) (revised SI Fig. 1) yields a tuning rate of 126.5 GHz/(MV/m). Now using the Lorentz local field approximation (see Lukin et al. NPJ Quantum Information 2020 [6])—where in silicon the local field is $4.56\times$ greater than the applied field—the predicted Stark tuning rate is thus 27.75 GHz/(MV/m). We note that this predicted polarizability is on the same order of the 4.5-35 GHz/(MV/m) observed with the divacancy in silicon carbide observed by Anderson et al. Science 2019 [1].

We opted not to include these estimates due to their failure to fully capture the background dopant-induced spatial distribution we (and others [1]) experimentally observe. For this reason, we continue to report our findings in terms of applied voltage, so that any other experimentalists performing similar measurements can use this as an accurate reference.

- Action Taken: We added the above discussion and estimate to the revised supplemental information.
- I wonder whether this electrical manipulation is reversible.
- Reply: We thank the reviewer for this excellent question, as it did not previously occur to us to discuss this interesting point. Our observed electrical manipulation is fully reversible and repeatable, as the various data sets presented in the main and supplemental text were repeatedly measured and over many weeks.
- Action Taken: We have added the following text to the manuscript to discuss this insightful question raised by the reviewer.

The center wavelength and nominal (zero-bias) brightness of the ensemble returns upon termination of the diode bias—both in the forward and reverse bias regimes—thus the charge state of the ensemble was not permanently altered by the measurements. Additionally, there is no observed time-delay in the restoration of the emitter optical properties, though more sensitive future time-domain measurements will provide precise detail on the transient nature of the response.

0.3 Reviewer #3 (Remarks to the Author):

The authors report electrical tuning of the optical properties of the ensemble G centers in Si diode devices. The wavelength tuning is of vital importance for the quantum information technologies. They fabricated the G centers with hydrogen implantation after carbon implantation. The fluorescence at 1278 nm appears only at the region treated with hydrogen implantation, demonstrating the position selectivity for the G center fabrication. The electrical tuning of the emission wavelength and intensity is reported for the G centers. However, I think the present results are insufficient for the publication in Nature Communications.

- Although the wavelength tuning is demonstrated, the intensity simultaneously decreases. This fact means that the formation of the depletion layer under voltage application makes the G centers unstable. Therefore, this technique is hard to directly use for the control of the G centers for quantum applications.
- Reply: We believe that the wavelength tuning is a result of the Stark shift interaction of the local electric field with the color centers. A *separate phenomenon* that also takes place within the depletion region, is the possibility of achieving charge populations of the inter-bandgap defect levels that are different from populations in non-depleted regions, affected by interactions with local electrons in the conduction band and holes in the valence band. It is known for other color center qubits that the charge value of the corresponding state can influence whether or not there is a radiative or non-radiative (dark) transition. An example is NV^- compared to NV^0 in diamond. The voltage-tuning, rather than making the G center *unstable*, may be placing it in a “dark” charge state. While this may not be desirable for the ultimate utilization of G centers in quantum systems, the important insight is that the G center *may have different charge states* that are “bright” or “dark”, and that a controlled charge population is required to maintain the G centers in the preferred state.

Additionally we note that the ability we have shown to convert a defect from an optically bright state to a dark one is similarly possible to occur in the reverse—where Fermi engineering via applied reverse bias may favorably populate a bright state relative to a dark one. The equilibrium Fermi level of the substrate will dictate the charge population of the defect, which is given by the background doping. In our demonstration with the host wafer we selected, G centers were optically active at equilibrium, and therefore increased reverse bias may have depopulated the optically active charge state.

- Action Taken: We have added the above discussion to the main text.
- The wavelength shift with 1.4 GHz/V seems to be large enough to obtain indistinguishable photons. The dependence of the tuning on the voltage (or electric field) should be more discussed in terms of the effect of the atomic symmetry of the G center and

electric dipole moment. Why isn't a single G center is utilized in this study to clarify these points?

- Reply: We agree with the reviewer that a detailed study of G center properties under electric field application would be valuable for the community. However, it is independently compelling to understand the behavior of G centers in engineered devices. The power of our technique lies in the insights to be learned about the behavior of *ensembles of color centers* in the presence of depletion regions and the associated electric fields, and in particular the spatial dependence of the emitter-diode interaction. These first experiments serve to establish general behavior, with the statistical validation provided by numerous color centers dispersed over a wide spatial region. As shown in the revised Figure 5, the Stark tuning of even small numbers of G centers varies dramatically, depending on whether those G centers are in fully depleted regions, or not. Future experiments, with sparser densities of G centers, and with geometries that might allow us to tune the direction of the electric field, could provide more information that would allow better understanding of the precise mechanism of electric field-induced tuning.
- Action Taken: We have added discussion to the main text to emphasize our motivation for investigating an ensemble rather than a single emitter.
- The mechanism of the decrease in the intensity should be discussed. Is the charge state changed in the depletion layer, or another reason?
- Reply: We thank the reviewer for this insightful question. We believe that the decrease of intensity is most likely due to a change in the charge state of the G center, resulting in a transition that is "dark", i.e., non-radiative in nature. With increasing reverse bias voltage across the junction, we change the electron occupancy of the defect states.
- Action Taken: We have incorporated additional information on our understanding of charge state conversion in the depletion region, and we added additional data (Fig. 5d) to further support this assertion.

References

- [1] Christopher P Anderson et al. "Electrical and optical control of single spins integrated in scalable semiconductor devices". In: *Science* 366.6470 (2019), pp. 1225–1230.
- [2] Denis R Candido and Michael E Flatté. "Suppression of the Optical Linewidth and Spin Decoherence of a Quantum Spin Center in a p-n Diode". In: *PRX quantum* 2.4 (2021), p. 040310. URL: <https://journals.aps.org/prxquantum/abstract/10.1103/PRXQuantum.2.040310>.

- [3] Kee-Joo Chang and DJ Chadi. “Theory of hydrogen passivation of shallow-level dopants in crystalline silicon”. In: *Physical review letters* 60.14 (1988), p. 1422.
- [4] Lorenzo De Santis et al. “Investigation of the stark effect on a centrosymmetric quantum emitter in diamond”. In: *Physical Review Letters* 127.14 (2021), p. 147402.
- [5] R.B. Hammond and R.N. Silver. “ANALYSIS OF LO AND TO PHONON ASSISTED FREE EXCITON LUMINESCENCE IN SILICON”. In: *Solid State Communications* 28 (1978), pp. 993–996. URL: <https://www.sciencedirect.com/science/article/abs/pii/0038109878906567>.
- [6] Daniil M Lukin et al. “Spectrally reconfigurable quantum emitters enabled by optimized fast modulation”. In: *npj Quantum Information* 6.1 (2020), p. 80.

REVIEWERS' COMMENTS

Reviewer #1 (Remarks to the Author):

The authors have replied all the concerns from reviewers, clarified the confusions and made necessary changes in the revised manuscript. It could be published as it is now.

Reviewer #2 (Remarks to the Author):

The authors have answered my questions and revised the manuscript to improve its quality. I thus recommend it to be published in Nature Communications.

Reviewer #3 (Remarks to the Author):

The manuscript has been revised according to the comments. Thus, I recommend the publication of the manuscript.